# Influence of Vitamin D Status and Supplementation on Metabolomic Profiles of Older Adults

**DOI:** 10.3390/metabo13020166

**Published:** 2023-01-23

**Authors:** Aislinn F. McCourt, Aifric M. O’Sullivan

**Affiliations:** UCD School of Agriculture and Food Science, UCD Institute of Food and Health, University College Dublin, D04 V1W8 Belfield, Ireland

**Keywords:** vitamin D, metabolomics, older adults

## Abstract

Metabolomics can identify metabolite patterns associated with different nutrition phenotypes and determine changes in metabolism in response to nutrition interventions. Vitamin D insufficiency is associated with increased metabolic disease risk; however, the role of vitamin D in metabolic health is not fully understood. This randomised, placebo-controlled trial (RCT) examined the influence of vitamin D status and the effect of vitamin D supplementation on metabolomic profiles in older adults. Healthy adults aged 50+ were randomly assigned to consume 20 µg vitamin D3 or a placebo daily for 4 weeks. Serum samples were collected at baseline and post-intervention for 25(OH)D and metabolomics analysis via liquid chromatography tandem mass spectrometry (LC-MS/MS). Pearson’s correlation examined relationships between 25(OH)D and metabolite concentrations. GLM ANCOVA compared metabolite concentrations between vitamin D-insufficient (<50 nmol/L) and -sufficient (>50 nmol/L) participants. The repeated-measures general linear model of covariance (RM GLM ANCOVA) examined changes in metabolites over time. Out of 132 metabolites, 2 short chain fatty acid concentrations were higher in the insufficient participants compared to sufficient participants, and 11 glycerophospholipid concentrations were lower in insufficient participants compared to sufficient participants at baseline. Three acylcarnitine concentrations decreased with vitamin D supplementation in vitamin D-insufficient participants. Our findings suggest that vitamin D status influences lipid metabolism in healthy older adults and supports the use of metabolomics in vitamin D research.

## 1. Introduction

Metabolomics involves the comprehensive systematic profiling of metabolites in a biofluid or tissue sample [1]. Nuclear magnetic resonance (NMR) and mass spectrometry (MS) combined with a gas or liquid chromatography separation method (GC-MS or LC-MS) are the most widely used metabolomic techniques [2,3]. Metabolomics is valuable in nutrition research, where metabolic signals are often weaker than pharmaceuticals, and metabolomic profiling can offer increased sensitivity compared to other analytical methods [4]. Metabolomics can identify food intake biomarkers, metabolite patterns associated with diets, or determine changes in metabolism in response to foods, supplements or diets [5,6,7]. Metabolomics is also used in nutrition research to describe the metabolic phenotype associated with a particular physiological state. For example, a recent systematic review describes higher concentrations of branched-chain amino acids, aromatic amino acids, lipids and acylcarnitines in metabolically unhealthy obese cohorts compared to metabolically healthy cohorts [8]. The authors suggest that the favourable metabolic profile associated with metabolically healthy obesity might explain the reduced disease risk for this cohort, and could be used as a way to monitor metabolic changes in both directions [8]. That is, the metabolomic profile could be used for early detection of a transition to metabolically unhealthy obesity or, conversely, could be used to monitor those who, as a result of some diet or lifestyle intervention, might move from unhealthy to healthy [8]. Thus, metabolomics can contribute to our understanding of how nutrition status impacts health, and can provide a means to monitor metabolic changes in response to a change in nutrition status.

The Institute of Medicine defines 25-hydroxyvitamin D (25(OH)D) concentrations >50 nmol/L, <50 nmol/L and <30 nmol/L as vitamin D sufficient, insufficient and deficient, respectively [9]. Vitamin D sufficiency is associated with reduced metabolic disease risk; however, the direct actions of vitamin D on metabolism are not fully understood [10,11,12]. For example, pre-clinical studies suggest that vitamin D protects β-cells from auto-immune destruction in type 1 diabetes [13]. However, despite associations between 25(OH)D and biomarkers of glucose metabolism in healthy and pre-diabetic cohorts, other mechanistic actions of vitamin D on glucose metabolism are not clear [10]. Metabolomics offers potential here and might help to characterise the impact of vitamin D on metabolic health, similarly to the application in metabolic healthy obese cohorts described earlier [8]. To date, very few published studies have examined metabolomic profiles associated with vitamin D status [14,15,16,17], and the heterogeneity between studies combined with relatively small sample sizes make it difficult to consolidate key findings. Other studies have reported metabolomic profiles following vitamin D interventions [7,18,19,20,21,22,23,24]. Again, there are limitations when attempting to consolidate findings; however, the evidence suggests that vitamin D supplementation impacts lipid metabolism, at least in unhealthy cohorts [19,20,21,22,23,24]. Conversely, studies with healthy cohorts have reported no change in metabolomic profiles with vitamin D supplementation [7,18].

Metabolomics is a powerful tool in human nutrition research; however, its applications in vitamin D research are limited. In particular, very few studies reporting metabolomic profiles related to vitamin D status or change in vitamin D status have recruited healthy adults, and only one has focused on healthy older adults specifically. We expect that metabolomics has more to offer with respect to understanding the role of vitamin D in metabolic health. Therefore, the aim of this study is to characterise vitamin D-insufficient and vitamin D-sufficient metabolomes and determine the effect of a 4-week vitamin D intervention on the metabolomic profiles of older adults.

## 2. Materials and Methods

### 2.1. Participants and Study Design

The University College Dublin (UCD) Human Research Ethics Committee (LS-19-69 -McCourt-O’Sullivan) granted ethical approval for the study. The study was conducted in line with the Declaration of Helsinki, and the trial was registered at clinicaltrials.gov (NCT04156074). Eighty-three community-dwelling healthy adults aged 50+ years, with no underlying diseases or conditions that required chronic medical or nutritional treatment, were recruited to the study from University College Dublin and surrounding areas in Dublin, Ireland. The intervention was completed in two phases between October 2019 and March 2020. Participants were excluded if they were taking a vitamin D supplement, if they had visited a country with high sun exposure in the past two months or if they planned to visit a country with high sun exposure during the intervention period. The study was conducted during the winter months in Ireland and excluded participants who were recently exposed to UVB sun rays in order to reduce the potential impact of cutaneous vitamin D synthesis on 25(OH)D concentrations during the study. Participants were randomly assigned to one of 4 intervention groups for 4 weeks. The intervention groups were as follows: vitamin D olive oil dairy drink, vitamin D coconut oil dairy drink, placebo control coconut oil dairy drink and vitamin D supplement. All groups randomised to receive vitamin D consumed 20 µg (800IU) vitamin D3 in a drink or supplement daily. Both participants and researchers were blinded to the dairy drink groups; however, the supplement group was not blinded. Seventy-four participants completed the study (Figure 1). Fasted serum samples and anthropometrics were collected at baseline and post-intervention. Compliance was measured via questionnaire as well as returned packaging and leftovers. The vitamin D olive oil dairy drink, vitamin D coconut oil dairy drink and vitamin D supplement groups were collapsed into one group, as there were no differences between the three vitamin D groups for any measurements at baseline or post-intervention. This collapsed group is referred to as the “vitamin D group” from here.

### 2.2. Dietary and Lifestyle Assessment

Habitual food and drink intake were assessed using a 4-day food diary. Habitual diet refers to the foods and drinks that people consume constantly or regularly over long periods. Participants recorded the amount, type and brand of all food, drinks, supplements and medications consumed during 4 consecutive days, including at least 1 weekend day, between baseline and the second visit in order to give an estimate of their habitual diet. While weights or portion sizes were not documented, participants clarified the quantity of food or drink consumed using standard portion sizes and household measures at visit 2. Researchers also probed for any other missing information. Dietary intake data were entered into Nutritics software (Nutritics Research Edition, v5.095, Dublin, Ireland) for analysis. All dietary data were quality controlled for accuracy by rechecking the foods and weights entered. Nutrient intake data were exported to IBM SPSS Statistics, version 24 (IBM Corp., Armonk, NY, USA) and mean daily nutrient intakes were calculated. Underreporters of energy intake were identified as having a ratio of energy intake to basal metabolic rate of <1.1, which was calculated using the Henry equation. Dietary intake data were collected from 75 of the 83 participants enrolled in the study. Dietary intake data were analysed for the full cohort, excluding underreporters (*n* = 7). There was no difference in the results from both groups; therefore, results are presented with underreporters included. Participants were collapsed into vitamin D intake groups, with low and high vitamin D consumers defined as consuming <5 µg (*n* = 35) and ≥5 µg vitamin D/day (*n* = 40) from their habitual diet and supplements without the study intervention, respectively.

### 2.3. Blood Sample Collection and Analysis

Fasted blood samples were collected by a trained phlebotomist into 10 mL clot activator serum tubes (BD Vacutainer, Dublin, Ireland). Each sample was inverted 5 times and clotted for 30 min at room temperature. Samples were centrifuged at 1500 RCF for 15 min at 20 °C (Rotina 38R, Hettich, France). After centrifugation, the samples were aliquoted and stored at −80 °C until analysis.

### 2.4. Serum Vitamin D Measurement

25(OH)D was measured as a vitamin D biomarker and was assessed by quantification of total 25(OH)D (D2 and D3) by a validated chromatography-tandem mass spectrometry (LC-MS/MS) method (Chromsystems Instruments and Chemicals GmbH) (API 4000; AB SCIEX, Macclesfield, UK). Analysis was performed in the Biochemistry Department of St James’s Hospital (accredited to ISO 15189) [25,26]. The quality and accuracy of the method was monitored by the use of internal quality controls, participation in the Vitamin D External Quality Assessment Scheme (DEQAS) and the use of the National Institute of Standards and Technology (NIST) 972 vitamin D standard reference material. The respective inter- and intra-assay coefficients of variation were 5.7% and 4.5% [26,27]. Participants were collapsed into vitamin D status groups, with 25(OH)D <50 nmol/L and ≥50 nmol/L defined as vitamin D-insufficient and -sufficient, respectively [9].

### 2.5. Biomarkers of Metabolic Health

Standard commercial kits measured biomarkers of metabolic health according to the manufacturer’s instructions. Serum glucose, total cholesterol (TC), high-density lipoprotein cholesterol (HDL-C), triglycerides (TG) and C-reactive protein (CRP) were measured using the Randox Daytona (Randox Laboratories, Antrim, UK). Low-density-lipoprotein cholesterol (LDL-C) was calculated using the Friedewald formula [28]: LDL-C = (TC-HDL-C)–(TG/2.17) mmol/L.

### 2.6. Metabolomics Analysis

Serum samples were analysed using a combination of direct injection mass spectrometry with a reverse-phase liquid chromatography (LC)-MS/MS custom assay, in combination with an ABSciex 4000 QTrap (Applied Biosystems/MDS SCIEX, Macclesfield, UK) mass spectrometer [29,30]. The method combines the derivatization and extraction of analytes, as well as the selective mass-spectrometric detection, using multiple reaction monitoring pairs. Isotope-labelled internal standards and other internal standards were used for metabolite quantification. The custom assay contained a 96 deep-well plate with a filter plate attached with sealing tape, and reagents and solvents were used to prepare the plate assay. First, 14 wells were used for 1 blank, 3 zero samples, 7 standards and 3 quality control samples. For all metabolites except organic acids, samples were thawed on ice and then vortexed and centrifuged at 13,000× *g*. Then, 10 µL of each sample was loaded onto the centre of the filter on the upper 96-well plate and dried in a stream of nitrogen. Subsequently, phenyl-isothiocyanate was added for derivatization. After incubation, the filter spots were dried again using an evaporator. Metabolites were then extracted by adding 300 µL of extraction solvent. The extracts were obtained by centrifugation into the lower 96-deep well plate, followed by a dilution step with MS running solvent.

For organic acid analysis, 150 µL of ice-cold methanol and 10 µL of isotope-labelled internal standard mixture were added to 50 µL of serum sample for overnight protein precipitation and centrifuged at 13,000× *g* for 20 min. Then, 50 µL of supernatant was loaded into the centre of wells of a 96-deep well plate, followed by the addition of 3-nitrophenylhydrazine reagent. After incubation for 2 h, butylated-hydroxytoluene stabilizer and water were added before LC-MS injection. Mass spectrometric analysis was performed using an ABSciex 4000 Qtrap tandem mass spectrometry instrument (Applied Biosystems/MDS Analytical Technologies, Foster City, CA, USA) equipped with an Agilent 1260 series ultra-high performance-LC system (Agilent Technologies, Palo Alto, CA, USA). The samples were delivered to the mass spectrometer by an LC method followed by a direct injection method. Data analysis was performed using Analyst version 1.6.2 (Applied Biosystems/MDS Analytical Technologies, Foster City, CA, USA). A total of 132 metabolites were quantified, including 39 acylcarnitines, 23 amino acids, 14 biogenic amines, 34 glycerophospholipids, 14 organic acids and 5 other metabolites. A full list of the metabolites measured are included in Appendix A.

### 2.7. Participant Flow and Data Collection

Participant enrolment, allocation, follow-up and analysis are described in the consort flow diagram (Figure 1). In total, 83 participants (40 males and 43 females) were enrolled in the study. Randomisation resulted in 65 participants being allocated to the vitamin D group (receiving 20 µg (800IU) vitamin D) and 18 were allocated to the placebo group. In total, 74 participants (37 males and 37 females) completed the study, 57 in the vitamin D group and 17 in the placebo group. Nine participants were lost to follow up or discontinued the intervention, either for personal reasons or reasons related to the COVID-19 pandemic. Data analysis and data presented in tables are for 74 participants. At baseline, 19 participants were classified as vitamin D-insufficient based on 25(OH)D <50 nmol/L, and 55 participants were classified as vitamin D-sufficient based on 25(OH)D >50 nmol/L. Baseline metabolomic data were compared between low (*n* = 35) and high (*n* = 40) habitual vitamin D intake groups as well. Dietary intake data were available for 75 participants; therefore, this analysis is based on *n* = 75, rather than *n* = 74.

### 2.8. Statistical Analysis

Statistical analysis was performed using IBM SPSS Statistics, version 24 (IBM Corp., Armonk, NY, USA) and RStudio version 4.0.3 (PBC, Boston, MA, USA). Shapiro–Wilk tests determined variable distribution, and outliers were examined using histograms. Any non-normal variables were transformed to normality using Johnson transformation. Data are presented as mean ± standard error of the mean (SE) or median and interquartile range. Univariate general linear model analysis of covariance (GLM ANCOVA) compared differences in participant characteristics between the vitamin D and placebo control intervention groups, differences in baseline and post-intervention metabolite concentrations between vitamin D status (vitamin D-insufficient vs. -sufficient) and vitamin D intake groups (low v high vitamin D intake) and the change in metabolite concentrations between the vitamin D and the placebo control intervention groups. Repeated measures (RM) GLM ANCOVA examined time*treatment interactions, as well as the simple main effects of time and treatment on metabolite concentrations following intervention. Baseline 25(OH)D was a significant covariate in the RM GLM ANCOVA; therefore, participants were split into vitamin D-insufficient and -sufficient groups for analysis. Sex, age, body mass index (BMI), body fat percentage, baseline 25(OH)D and nutrient intakes were considered as potential covariates, and any significant covariate (*p* < 0.05) was included in the final ANCOVA models. Pearson’s correlation explored relationships between 25(OH)D and metabolites or changes in metabolites at baseline and post-intervention. The Benjamini–Hochberg false discovery rate (FDR) correction, grouped by metabolite type, was applied to all results to account for multiple testing.

## 3. Results

### 3.1. Participant Characteristics

Characteristics of vitamin D-insufficient (<50 nmol/L) and -sufficient (≥50 nmol/L) participants who completed the study (*n* = 74) are presented in Table 1. Participants had a mean age of 60 ± 8 kg and BMI of 28.1 ± 0.9 kg/m^2^ at baseline, and there were no differences in biomarkers of metabolic health between those randomised to receive either vitamin D or the placebo control. Mean daily vitamin D intake from food alone at baseline was 4.7 ± 0.5 µg. Mean baseline 25(OH)D concentration was 60.3 ± 2.6 nmol/L. Intervention compliance was high, with a median compliance of 28 (26–29) days or 100.0 (96.0–100)%. There was a significant increase in 25(OH)D concentrations in the vitamin D group after intervention, with a mean 25(OH)D change of 10.7 ± 2.1 nmol/L compared to −3.9 ± 1.4 nmol/L in the placebo control group (Table 1). Fourteen participants in the vitamin D group were vitamin D-insufficient at baseline, but this number decreased to three post-intervention. There were no changes in biomarkers of metabolic health (Table 1).

### 3.2. Baseline Vitamin D and Metabolomic Profiles

Table 2 presents metabolites that were significantly different between vitamin D-insufficient and -sufficient participants, as well as significant correlations between 25(OH)D concentrations and metabolites at baseline. After FDR correction, butyric and isobutyric acid concentrations were significantly higher in insufficient participants compared to sufficient participants; and 11 glycerophospholipids were lower in vitamin D-insufficient participants compared to vitamin D-sufficient participants (Table 2). Table 3 presents significant differences in metabolite concentrations between high and low vitamin D consumers, as well as significant correlations between baseline metabolite concentrations and vitamin D intake.

### 3.3. Effect of Vitamin D Supplementation on Metabolomic Profiles

The intervention effects on metabolite concentrations were examined using RM GLM ANCOVA. Table 4 presents any significant time*treatment, time and treatment effects on metabolites in vitamin D-insufficient and -sufficient participants. In the insufficient participants, there were no significant time*treatment or time effects on metabolites after FDR correction (Table 4). However, there was a simple main effect of treatment on three acylcarnitines after FDR correction (q < 0.05), with higher C14.1OH, C16.2 and C16.1 concentrations in the vitamin D group compared to the placebo control group (Table 4). In the vitamin D-sufficient participants, there were no significant time*treatment, time or treatment effects on metabolites after FDR correction (Table 4).

Associations between baseline 25(OH)D and post-intervention metabolites were examined (Table 5). LYSOC26.0 concentrations were negatively associated with baseline 25(OH)D; however, this was not significant after FDR correction (Table 5). Next, associations between post-intervention metabolites and post-intervention 25(OH)D concentrations were examined (Table 5). Post-intervention 25(OH)D was negatively associated with glutamic acid and positively associated with methionine sulfoxide after FDR correction (Table 5). Associations between baseline 25(OH)D and changes in metabolite concentrations were then examined (Table 5). A change in CD.1DC was positively associated with baseline 25(OH)D after FDR correction (Table 5). Lastly, associations between change in 25(OH)D and change in metabolites were examined; however, there were no significant correlations after FDR correction.

## 4. Discussion

To the best of our knowledge, this is the first study comparing metabolomic profiles of vitamin D-insufficient (<50 nmol/L) and -sufficient (≥50 nmol/L) healthy older adults. This is also the largest study examining the effect of vitamin D supplementation on metabolomic profiles in older adults. There were some differences in the metabolomic profiles of vitamin D-insufficient and -sufficient participants at baseline, mainly in lipid metabolites; however, there were very minor changes in metabolomic profiles after a 4-week vitamin D intervention, despite a significant increase in 25(OH)D concentrations and a shift from insufficient to sufficient vitamin D status.

Firstly, taking an observational approach, we characterised vitamin D-insufficient and -sufficient metabolomes and examined differences in metabolite concentrations between these cohorts. A small number of metabolites were significantly different between the vitamin D status groups. Two SCFA concentrations were higher in the insufficient participants compared to the sufficient participants, and eleven glycerophospholipid concentrations were lower in insufficient participants compared to sufficient participants. Previous studies examined associations between 25(OH)D concentrations and metabolites in diverse cohorts, and also reported associations with lipid metabolites [14,15,16,17]. While there is evidence that vitamin D plays a role in lipid metabolism (for example, through its role as an inhibitor of sterol regulatory element-binding protein (SREBP) activation), very few studies have reported data from human studies [31]. Leung et al. performed one of the largest exploratory studies, examining relationships between 25(OH)D and metabolomic profiles, and reported 25 positive and 36 negative correlations between lipids and 25(OH)D, including 7 negative associations between phospholipids and 25(OH)D [4]. It is unclear why we observed higher phospholipid concentrations in vitamin D-sufficient participants and why Leung et al. reported negative associations between 25(OH)D and phospholipids. However, our results are supported by older animal studies reporting that vitamin D status alters phospholipid metabolism and the phospholipid composition of cell membranes [32,33]. One of these animal studies reported that vitamin D-deprived rats had lower renal cell membrane phosphatidylcholine and phosphatidylethanolamine concentrations, which may explain the positive association reported here [32]. Again, the exact link between higher SCFA and vitamin D insufficiency is unclear. However, there is evidence to suggest a link between higher circulating 1,25(OH)2D concentrations and higher butyrate-producing bacteria in the gut microbiome [34]. Although we did not report 1,25(OH)D concentrations herein, this does suggest a potential link between vitamin D status and SCFAs. Despite evidence linking vitamin D status and lipid metabolites, the relatively small number of studies and inherent diversity substantiate the need for future research examining the underlying mechanisms that may be driving differences in lipid metabolites between vitamin D status groups.

The second part of this research examined how metabolites changed in response to a 4-week vitamin D intervention compared to a placebo control. However, we reported only three significant metabolite changes in response to vitamin D supplementation and three associations between baseline 25(OH)D concentrations and metabolite changes with supplementation. It should be noted that despite variation in response, the 25(OH)D concentrations of all participants randomised to the vitamin D group increased, and all except three moved from vitamin D-insufficient to vitamin D-sufficient categories over the course of the study. Therefore, the fact that metabolite concentrations did not change is likely not a function of a lack of response to intervention. To the best of our knowledge, only three other studies have reported metabolomic profiles before and after a vitamin D intervention with healthy participants, and all three reported no change in metabolomic profiles with treatment [7,18,20]. One of these studies was very similar to the current study, except that it had a smaller sample size [18]. Another focused primarily on the impact of vitamin supplementation in patients with cystic fibrosis; however, it also reported no effect of treatment in the healthy control group [20]. In contrast, studies in compromised cohorts appear to have reported significant changes in metabolomic profiles with vitamin D treatment [19,20,21,22,23,24]. For example, pathway enrichment analysis in a vitamin D-insufficient cohort supplemented with 15, 100 or 250 µg vitamin D/day for 24 weeks showed changes in the metabolites involved in lipid oxidation [22]. This effect is not likely driven by the higher doses administered compared to our study, as there were no differences between the three treatment groups [22]. Although we reported no change in the metabolome after vitamin D supplementation, even for those who were vitamin D-insufficient at baseline, it is possible that we might see a similar effect to that reported by Shirvani and colleagues [22] with a larger insufficient cohort. Lastly, an effect of vitamin D on lipid metabolites was also reported in another long intervention in participants with obesity [21]. In the 4-month intervention, 100 µg vitamin D daily altered a small number of lipid metabolites in vitamin D-insufficient participants with metabolically unhealthy obesity, but not in those who were metabolically healthy [21]. While we had participants with both vitamin D insufficiency and obesity, it is possible that we did not see an effect in these participants due our smaller sample size. Therefore, based on these two studies, it is possible that longer interventions are needed to detect changes in blood metabolites, as they are subject to strict homeostasis. However, the results of intervention studies are currently non-conclusive due to the heterogeneity of participant types, doses and study duration. Due to this heterogeneity, it is difficult to determine whether there is a true impact of vitamin D on metabolomic profiles, or if the results are incidental.

While metabolomics has provided some new insights into links between vitamin D and human metabolism, several questions remain. Evidence from observational and intervention studies suggests that vitamin D affects lipid metabolism; however, the small number of studies, the differences in study design and the relatively small sample sizes make it difficult to draw major conclusions. It is also important to note that blood metabolites are tightly controlled within a homeostatic range; therefore, longer interventions might show more changes in metabolites in response to changes in circulating 25(OH)D concentrations. Lastly, metabolomics studies that recruit more participants and take vitamin D status into account at the recruitment stage are needed so that changes according to baseline status can be examined in more detail. It is clear that there is more to accomplish to explore links between vitamin D and metabolic health, and we maintain that metabolomics holds potential in this area. In addition, radiolabelling has not yet been used in vitamin D metabolomics studies, but has been used previously in combination with metabolomics in different interventions to provide a comprehensive understanding of metabolic pathways [35]. Radiolabeled vitamin D would track vitamin D through metabolism and could elucidate the role of vitamin D in certain metabolic pathways. Therefore, the use of metabolomics in vitamin D research is still in its infancy. The current small body of research can inform comprehensive methodologic approaches to conduct and reproduce large-scale studies in order to contribute to our understanding of vitamin D in metabolism.

There are inherent strengths and limitations to consider when interpreting these results. Firstly, with respect to the strengths, 25(OH)D and metabolite concentrations were quantified by LC-MS/MS using standardised methods. In Ireland, vitamin D can be synthesised cutaneously from April to September. Therefore, the influence of cutaneous vitamin D synthesis was minimised, as the study was carried out between November to March and participants were excluded if they were traveling abroad during the study period. We also examined metabolomic profile differences between high and low vitamin D consumers, and other studies that did not account for vitamin D intake identified this as a limitation of their analyses [4]. Most importantly, this is secondary analysis of data collected as part of an RCT powered to detect a change in 25(OH)D in response to vitamin D supplementation relative to the control. We included metabolomics analysis as an exploratory dataset as part of this research with an aim to identify more subtle metabolic changes that could potentially occur in response to a change in 25(OH)D concentrations. As such, this exploratory study is hypothesis-generating and can be used to power future studies in healthy older adults. Additionally, participant numbers were reduced due to the COVID-19 pandemic, and future studies should aim to recruit a larger number of participants. Lastly, there was significant variation in baseline 25(OH)D concentrations, resulting in a large distribution of baseline 25(OH)D concentrations in each intervention group, even when splitting the groups into insufficient and sufficient.

Vitamin D sufficiency is essential for overall health. We reported higher SCFA concentrations in vitamin D-insufficient participants, suggesting a role of vitamin D in SCFA metabolism in older adults. In addition, concentrations of glycerophospholipids were lower in vitamin D-insufficient participants compared to vitamin D-sufficient participants, suggesting a potential role of vitamin D in lipid metabolism and, thus, metabolic health. However, research to date is too limited to determine mechanistic actions of vitamin D status on lipid metabolites. This is the largest study examining the effect of vitamin D supplementation on metabolite concentrations in healthy older adults. We observed no effect of a 4-week vitamin D intervention on metabolite concentrations; however, this study is a secondary analysis, and was not powered to detect these changes. While research suggests that results may differ in cohorts with a larger proportion of obesity and vitamin D insufficiency, this study can power future research in healthy older adults. To conclude, our findings suggest an influence of vitamin D status on lipid metabolism in healthy older adults, and also support the use of metabolomics in vitamin D interventions.

## Figures and Tables

**Figure 1 metabolites-13-00166-f001:**
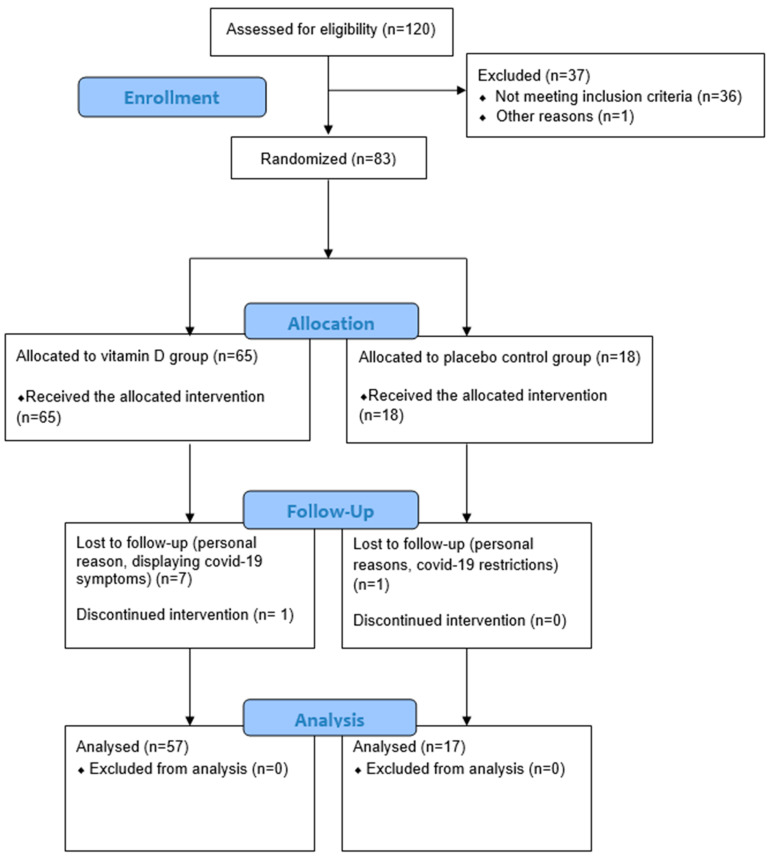
Flowchart of study progression.

**Table 1 metabolites-13-00166-t001:** Participant characteristics, including age, BMI, 25(OH)D and biomarkers of metabolic health.

	Vitamin D-Insufficient (<50 nmol/L)					Vitamin D-Sufficient (≥50 nmol/L)				
	Vitamin D		Placebo Control			Vitamin D		Placebo Control		
	*n* = 14		*n* = 5			*n* = 43		*n* = 12		
	Mean	SE	Mean	SE	*p*-Value	Mean	SE	Mean	SE	*p*-Value
Age (years)	58.0	1.7	57.6	2.9	0.88	61.9	1.3	61.3	2.3	0.81
BMI (kg/m^2^)	29.0	0.7	27.6	1.2	0.36	28.3	0.5	27.5	0.8	0.38
BL 25(OH)D (nmol/L)	33.8	2.5	38.9	4.3	0.33	71.4	0.3	70.4	4.0	0.83
Post 25(OH)D (nmol/L)	60.1	2.9	37.3	5.0	0.001	75.0	2.0	66.8	3.5	0.05
BL TC (mmol/L)	4.9	0.2	4.3	0.3	0.06	5.1	0.2	5.6	0.3	0.15
Post TC (mmol/L)	5.0	0.2	4.3	0.3	0.06	5.1	0.2	5.7	0.3	0.11
BL HDL (mmol/L)	1.6	0.1	1.4	0.1	0.14	1.6	0.1	1.8	0.1	0.13
Post HDL (mmol/L)	1.6	0.1	1.4	0.1	0.22	1.7	0.1	1.8	0.1	0.37
BL LDL (mmol/L)	2.7	0.2	2.4	0.3	0.30	3.0	0.2	3.1	0.3	0.62
Post LDL (mmol/L)	2.7	0.2	2.4	0.3	0.26	3.0	0.2	3.3	0.3	0.34
BL TRIG (mmol/L)	1.3	0.2	1.2	0.4	0.78	1.2	0.1	1.3	0.2	0.35
Post TRIG (mmol/L)	1.3	0.2	1.3	0.3	0.88	1.1	0.1	1.3	0.2	0.36
BL GLUC (mmol/L)	5.3	0.2	5.8	0.4	0.85	5.4	0.1	5.5	0.1	0.55
Post GLUC (mmol/L)	5.3	0.2	5.7	0.4	0.46	5.2	0.2	6.3	0.4	0.027
BL CRP (mmol/L)	2.3	0.5	1.5	0.8	0.37	2.2	0.3	1.8	0.6	0.60
Post CRP (mmol/L)	2.5	0.6	1.5	1.1	0.44	2.0	0.3	1.8	0.5	0.75

Data are presented as mean ± standard error. GLM ANCOVA, controlling for sex, body fat percentage, BMI and baseline 25(OH)D, explored differences between groups. *p* < 0.05 was considered statistically significant. SE, standard error; BL, baseline, 25(OH)D, 25-hydroxyvitamin D, Post, post-intervention.

**Table 2 metabolites-13-00166-t002:** Metabolite concentrations in vitamin D-insufficient and -sufficient participants and correlations between 25(OH)D concentrations and metabolites at baseline.

	Insufficient		Sufficient						
	(<50 nmol/L)		(≥50 nmol/L)						
	*n* = 19		*n* = 55		GLM ANCOVA		Pearson’s Correlation		
Metabolite (µM)	Mean	SE	Mean	SE	*p*-Value	q-Value	r	*p*-Value	q-Value
**Amino acids**									
Threonine	105.6	5.2	98.4	3.0	0.22	0.61	−0.260	0.039	0.43
Leucine	119.3	4.9	108.8	3.1	0.05	0.60	−0.252	0.047	0.26
Isoleucine	65.5	2.8	57.3	1.6	0.005	0.11	−0.283	0.025	0.54
Methionine	23.3	0.9	21.8	0.4	0.12	0.37	−0.271	0.031	0.15
**Biogenic Amines**									
Acetyl-ornithine	0.767	0.077	1.048	0.127	0.12	0.81	0.359	0.004	0.027
Trans-hydroxy-proline	8.53	0.62	7.30	0.45	0.07	1.00	−0.386	0.002	0.025
**Histidines**									
Methylhistidine	8.64	1.14	12.08	1.05	0.044	0.24	0.034	0.79	0.91
**Short chain fatty acids**									
Butyric acid	1.074	0.061	0.852	0.042	0.003	0.033	−0.293	0.020	0.15
Propionic acid	0.978	0.049	0.841	0.029	0.018	0.13	−0.266	0.035	0.19
Isobutyric acid	1.321	0.070	1.062	0.040	0.002	0.044	−0.370	0.003	0.06
Hippuric acid	5.67	0.67	7.06	0.74	0.44	0.56	0.301	0.016	0.18
**Glycerophospholipids**									
LYSOC16.0	46.1612	2.2421	53.0679	1.5080	0.023	0.05	0.334	0.007	0.08
LYSOC18.1	12.2015	0.7658	13.7980	0.4883	0.14	0.18	0.254	0.045	0.12
LYSOC18.0	13.3603	0.6416	15.1602	0.5183	0.08	0.12	0.300	0.017	0.08
LYSOC24.0	0.0764	0.0044	0.0850	0.0025	0.07	0.11	0.273	0.030	0.09
LYSOC26.1	0.0644	0.0041	0.0793	0.0028	0.008	0.030	0.334	0.007	0.06
LYSOC26.0	0.3970	0.0197	0.5063	0.0166	0.001	0.017	0.334	0.007	0.05
LYSOC28.1	0.2567	0.0116	0.3179	0.0134	0.022	0.05	0.231	0.07	0.15
LYSOC28.0	0.4684	0.0285	0.5402	0.0183	0.042	0.08	0.230	0.07	0.15
X14.1SMOH	5.9577	0.1863	7.0782	0.2366	0.019	0.05	0.186	0.14	0.23
X16.0SM	99.2598	2.9551	111.2394	2.5620	0.010	0.034	0.234	0.06	0.16
PC36.0AE	1.2157	0.0385	1.5191	0.0448	<0.001	<0.001	0.337	0.007	0.24
PC36.6AA	0.7877	0.0501	1.0911	0.0557	0.003	0.020	0.190	0.14	0.41
PC36.0AA	6.0304	0.2371	7.2132	0.2011	0.002	0.017	0.279	0.027	0.07
X22.2SMOH	12.7172	0.4626	14.5825	0.4303	0.024	0.05	0.172	0.18	0.28
PC38.6AA	58.3369	4.1387	79.2220	3.5197	0.001	0.011	0.295	0.019	0.07
PC38.0AA	2.6341	0.1392	3.3977	0.1397	0.003	0.017	0.296	0.019	0.16
PC40.6AE	3.5294	0.1794	4.5342	0.1984	0.006	0.026	0.274	0.030	0.09
X24.1SMOH	2.5160	0.1159	2.8307	0.0783	0.039	0.08	0.198	0.12	0.21
PC40.6AA	15.9087	1.0491	20.7540	0.9707	0.011	0.034	0.221	0.08	0.16
PC40.2AA	0.3716	0.0168	0.4461	0.0182	0.018	0.05	0.315	0.012	0.07
PC40.1AA	0.2617	0.0118	0.3296	0.0143	0.003	0.015	0.335	0.007	0.12
**Acylcarnitines**									
C3OH	0.0245	0.0013	0.0277	0.0007	0.028	0.16	0.134	0.29	0.84
C6	0.0732	0.0041	0.0887	0.0043	0.035	0.18	0.143	0.26	0.87
C5OH	0.0323	0.0012	0.0355	0.0008	0.024	0.16	0.206	0.11	1.00
C5.1DC	0.0137	0.0005	0.0153	0.0004	0.010	0.13	0.299	0.017	0.35
C9	0.0381	0.0047	0.0517	0.0036	0.005	0.20	0.317	0.011	0.46
C12	0.0918	0.0098	0.1204	0.0079	0.017	0.14	0.206	0.11	0.87
C14	0.0366	0.0031	0.0474	0.0026	0.011	0.11	0.192	0.13	1.00
C16	0.1045	0.0064	0.1243	0.0044	0.007	0.14	0.188	0.14	0.93

Data are presented as mean ± standard error. GLM ANCOVA, controlling for sex and BMI, explored differences in metabolite concentrations between vitamin D status groups. Pearson’s correlation examined relationships between baseline 25(OH)D concentrations and metabolite concentrations. *p* and q < 0.05 were considered statistically significant.

**Table 3 metabolites-13-00166-t003:** Metabolite concentrations in low and high vitamin D consumers and correlations between vitamin D intake and metabolites at baseline.

Low Consumers (<5 µg/day)			High Consumers (≥5 µg/day)						
	*n* = 35		*n* = 40		GLM ANCOVA		Pearson’s Correlation		
Metabolite (µM)	Mean	SE	Mean	SE	*p*-Value	q-Value	r	*p*-Value	q-Value
**Amino Acids**									
Alanine	340	10	378	11	0.038	0.84	0.166	0.15	1.00
Betaine	34.8	1.3	36.2	1.6	0.37	0.82	0.238	0.040	0.88
**Biogenic Amines**									
Trans-hydroxy-proline	7.64	0.60	7.29	0.47	0.43	0.76	0.248	0.032	0.45
**Short chain fatty acids**									
Butyric acid	1.026	0.061	0.796	0.036	0.001	0.022	−0.181	0.12	0.44
Propionic acid	0.945	0.037	0.807	0.031	0.001	0.011	−0.192	0.10	0.72
Isobutyric acid	1.218	0.051	1.037	0.049	0.002	0.015	−0.221	0.06	1.00
**Organic acids**									
Methylmalonic acid	0.2075	0.0245	0.1347	0.0095	0.007	0.039	−0.195	0.09	1.00
**Glycerophospholipids**									
LYSOC17.0	0.9392	0.0399	1.0656	0.0544	0.020	0.10	0.292	0.011	0.38
X14.1SMOH	6.2794	0.2000	7.2237	0.2910	0.004	0.14	0.109	0.35	0.67
X16.1SMOH	3.8978	0.1376	4.3241	0.1696	0.027	0.11	0.142	0.22	0.70
PC36.6AA	0.9254	0.0641	1.1019	0.0626	0.010	0.07	0.227	0.05	0.42
X22.2SMOH	13.4302	0.4098	14.8973	0.5612	0.006	0.10	0.169	0.15	0.62
PC38.6AA	67.2424	3.7674	82.0791	4.5278	0.016	0.09	0.207	0.07	0.51
PC40.6AE	3.9610	0.1998	4.6367	0.2450	0.009	0.08	0.254	0.028	0.48
PC40.6AA	17.4317	0.9263	21.9442	1.2138	0.007	0.08	0.227	0.05	0.56
**Acylcarnitines**									
C3	0.2505	0.0138	0.2859	0.0117	0.015	0.30	0.061	0.61	1.00
C4	0.1725	0.0100	0.2119	0.0148	0.016	0.21	0.050	0.67	1.00
C9	0.0412	0.0035	0.0534	0.0045	0.005	0.20	0.083	0.48	1.00

Data are presented as mean ± standard error. GLM ANCOVA, controlling for sex and BMI, was used to explore differences between vitamin D intake groups. Pearson’s correlation explored relationships between vitamin D intake and metabolite concentrations. *p* and q < 0.05 were considered statistically significant.

**Table 4 metabolites-13-00166-t004:** Effect of vitamin D supplementation on metabolite concentrations in vitamin D-insufficient and -sufficient participants.

	Insufficient (<50 nmol/L)							Sufficient (≥50 nmol/L)						
			*n* = 19							*n* = 55				
	Baseline		Post-Intervention		RM ANCOVA (q-Value)			Baseline		Post-Intervention		RM ANCOVA (q-Value)		
Metabolite (µM)	Mean	SE	Mean	SE	Time	Treatment	T * T	Mean	SE	Mean	SE	Time	Treatment	T * T
**Amino Acids**														
Leucine	119.3	4.9	122.3	5.9	0.33	0.83	1.00	108.8	3.1	117.5	3.5	1.00	1.00	1.00
Betaine	34	2	36.5	1.9	0.26	0.68	1.00	36.1	1.2	36.1	1.3	1.00	0.82	0.10
Taurine	62.8	4.3	67.9	4.2	0.92	0.98	0.53	66.4	2.5	67.0	2.2	1.00	0.85	0.29
Biogenic Amines														
Serotonin	1.021	0.106	1.237	0.166	0.99	1.00	0.76	0.894	0.063	0.986	0.072	1.00	1.00	0.24
Spermidine	0.238	0.008	0.242	0.007	1.00	1.00	0.14	0.241	0.004	0.234	0.004	1.00	1.00	0.77
Organic Acids														
Choline	9.36	0.44	9.16	0.37	0.24	1.00	1.00	9.48	0.28	9.34	0.24	1.00	1.00	0.80
**SCFA**														
Butyric acid	1.074	0.061	0.934	0.053	0.27	1.00	1.00	0.852	0.042	0.96	0.052	1.00	1.00	0.94
Propionic acid	0.978	0.049	0.97	0.057	0.22	1.00	0.95	0.841	0.029	1.032	0.036	1.00	0.98	0.98
Isobutryic acid	1.321	0.07	1.074	0.063	0.31	1.00	0.94	1.062	0.04	1.111	0.041	1.00	1.00	0.74
**Glycerophospholipids**														
LYSOC18.1	12.2015	0.7658	13.6343	0.9711	1.00	0.89	1.00	13.798	0.4883	14.3698	0.5167	0.96	0.78	0.79
LYSOC20.3	0.878	0.0834	1.0139	0.1067	1.00	0.85	0.69	0.8636	0.0568	1.0213	0.0578	0.99	1.00	0.17
PC32.2AA	5.8126	0.1846	5.8335	0.3155	1.00	0.71	0.98	6.2932	0.2394	6.0296	0.2074	1.00	0.73	0.85
**Acylcarnitines**														
C0	36.2855	1.4732	35.161	1.7376	1.00	0.66	0.97	36.2065	1.1691	35.8572	1.0347	1.00	0.32	0.49
C4OH	0.0421	0.0033	0.037	0.0023	0.96	0.51	0.96	0.0462	0.0031	0.0393	0.0018	1.00	0.29	0.53
C9	0.0381	0.0047	0.0356	0.004	1.00	0.65	0.96	0.0517	0.0036	0.0503	0.0038	1.00	0.33	0.34
C7DC	0.0651	0.0126	0.05	0.0123	1.00	0.86	0.53	0.0708	0.0076	0.0559	0.0074	1.00	0.31	0.79
C10 2	0.0615	0.0036	0.0441	0.0026	1.00	0.81	0.50	0.0593	0.0021	0.048	0.0019	1.00	0.73	0.78
C12.1	0.1112	0.0083	0.094	0.0071	1.00	0.06	0.97	0.1331	0.007	0.1097	0.0063	1.00	0.25	1.00
C12	0.0918	0.0098	0.0741	0.0063	1.00	0.06	1.00	0.1204	0.0079	0.0978	0.008	1.00	0.27	1.00
C14.2	0.0518	0.0051	0.0382	0.0032	1.00	0.06	1.00	0.0513	0.0033	0.0454	0.0033	1.00	0.24	0.78
C14.1	0.146	0.0131	0.1082	0.0092	1.00	0.05	0.97	0.1664	0.0102	0.1335	0.009	1.00	0.98	1.00
C14	0.0366	0.0031	0.0315	0.0025	1.00	0.06	1.00	0.0474	0.0026	0.0384	0.0024	1.00	0.28	0.72
C12DC	0.0066	0.0003	0.0062	0.0002	1.00	0.45	1.00	0.007	0.0002	0.0062	0.0002	1.00	0.48	0.77
C14.1OH	0.0198	0.0016	0.0174	0.0012	1.00	0.020	1.00	0.0214	0.0009	0.02	0.0009	1.00	0.45	0.74
C16.2	0.0129	0.0009	0.0118	0.0008	1.00	<0.001	1.00	0.0135	0.0007	0.0127	0.0006	1.00	0.25	0.83
C16.1	0.0453	0.0028	0.0435	0.0026	1.00	0.040	0.96	0.0515	0.0022	0.0465	0.002	1.00	0.25	0.82
C16	0.1045	0.0064	0.101	0.0068	1.00	0.11	1.00	0.1243	0.0044	0.1108	0.0037	1.00	0.31	0.71
C16.2OH	0.009	0.0004	0.0099	0.0004	1.00	0.11	1.00	0.0097	0.0003	0.0098	0.0003	1.00	0.25	0.75
C16.1OH	0.0141	0.0006	0.0137	0.0008	1.00	0.13	1.00	0.0146	0.0005	0.0144	0.0003	1.00	0.29	0.58
C16OH	0.0082	0.0003	0.009	0.0004	1.00	0.19	0.81	0.0086	0.0003	0.0093	0.0003	1.00	0.29	0.76
C18.2	0.0592	0.0037	0.0547	0.0037	1.00	0.17	0.78	0.0552	0.0021	0.0521	0.0021	1.00	0.25	0.55
C18.1	0.1532	0.0085	0.1424	0.0096	1.00	0.06	1.00	0.1619	0.0062	0.143	0.0057	1.00	0.42	0.82
C18	0.0431	0.0029	0.0385	0.0024	1.00	0.42	0.96	0.0504	0.0022	0.0419	0.0014	1.00	0.99	0.70
C18.1OH	0.0114	0.0005	0.0128	0.0004	1.00	0.07	1.00	0.0117	0.0004	0.0128	0.0004	1.00	0.35	0.24

Data are presented as mean ± standard error. RM GLM ANCOVA, controlling for sex and BMI, explored the effect of time*treatment, time and treatment in vitamin D status groups. q < 0.05 was considered statistically significant. SE, standard error; RM GLM ANCOVA, repeated-measures general linear model analysis of covariance; T*T, time*treatment; BL, baseline; Post, post-intervention.

**Table 5 metabolites-13-00166-t005:** Correlations between 25(OH)D concentrations and metabolite concentrations (*n* = 74).

Metabolite	r	*p*-Value	q-Value
**Baseline 25(OH)D and post-intervention metabolite concentrations**			
**Glycerophospholipids**			
LYSOC26.0	−0.274	0.018	0.61
**Post-intervention 25(OH)D and metabolite concentrations post-intervention**			
**Amino acids**			
Glycine	0.291	0.021	0.12
Taurine	0.294	0.019	0.14
Glutamic acid	−0.410	0.001	0.019
Citrulline	0.337	0.007	0.08
**Biogenic Amines**			
Putrescine	0.325	0.009	0.13
Methionine sulfoxide	0.344	0.006	0.040
Acetyl-ornithine	0.314	0.012	0.06
**Organic acids**			
HPHPA	0.266	0.035	0.39
Succinic acid	0.352	0.005	0.10
Methylmalonic acid	0.251	0.047	0.26
Homovanillic acid	0.265	0.036	0.26
**Carbohydrates**			
Glucose	−0.263	0.037	1.00
**Glycerophospholipids**			
PC36.0AA	0.333	0.008	0.26
PC38.0AA	0.303	0.016	0.18
PC40.2AA	0.270	0.032	0.27
PC40.1AA	0.323	0.010	0.17
**Acylcarnitines**			
C3.1	−0.264	0.036	1.00
C12	0.249	0.049	0.98
**Baseline 25(OH)D and change in metabolite concentrations**			
**Glycerophospholipids**			
LYSOC16.1	−0.239	0.040	0.28
LYSOC16.0	−0.332	0.004	0.13
LYSOC17.0	−0.251	0.031	0.26
LYSOC18.0	−0.268	0.021	0.24
LYSOC26.0	−0.298	0.010	0.17
**Acylcarnitines**			
C5.1DC	0.243	0.037	0.019
**Change in 25(OH)D and change in metabolite concentrations**			
**Amino Acids**			
Tryptophan	0.230	0.048	1.00
**Glycerophospholipids**			
LYSOC16.1	0.269	0.020	0.14
LYSOC16.0	0.338	0.003	0.06
LYSOC17.0	0.249	0.032	0.16
LYSOC18.0	0.309	0.007	0.08
LYSOC20.3	0.264	0.023	0.13
LYSOC26.0	0.356	0.002	0.06
PC40.6AA	0.271	0.020	0.17
**Acylcarnitines**			
C3OH	0.275	0.018	0.70

25(OH)D, 25-hydroxyvitamin D; r, Pearson’s correlation coefficient. *p* and q < 0.05 were considered statistically significant.

## Data Availability

The data presented in this study are available upon request from the corresponding author. The data are not publicly available due to ethical approval requirements.

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
