# Peer review of "Influence of Vitamin D Status and Supplementation on Metabolomic Profiles of Older Adults"

_metabolites, 2023, doi:10.3390/metabo13020166_

Round 1
Reviewer 1 Report
In this paper the authors describe the associations of vitamin D status, habitual vitamin D intake and a 4 week vitamin D supplementation with metabolic profiles of healthy older adults.
In general, this can be interesting to readers, but there are some issues with this study and manuscript that need to be addressed before publishing:
- Major issue: Number of participants: in the materials and methods section there is no information on the number of participants that were recruited, that completed the study, and that were included in the calculation and presentation of results, and based on which reasons. Although in the results section, it says that 63 participants completed the study, in the results presented in Tables 1 - 4 the numbers are inconsistent and are always given differently! The total numbers of participants vary between 73 and 82. The numbers for "insufficient" vary between 19 and 65, the number for "sufficient" vary between 17 and 55. This needs to be corrected or explained!
- Major issue: At the end of the discussion section as a limitation of the study it says "the study was not powered to detect changes in metabolomic profiles". This immediately triggers the question: So what's the point? Why did you perform the study anyway? The title of the paper indicates that this was the aim of the study. So how do you want the reader to interpret the results? In what way could this study generate hypotheses? Maybe it needs a bit more explanation what the intention was, why the study is underpowered, or whether this was a result of the statistical analyses to find that the number of participants was not sufficient...
- Minor issue: Throughout the text, it is difficult for the reader to know if "vitamin D status" or "vitamin D intake" refers to baseline or post-intervention. This takes a lot of time to try and guess what is referred to and makes reading very strenuous. This should be made more clear to make the text better legible.
- In the discussion section l. 290 - 292: This sentence makes no sense to me: Why should 1,25(OH)2D be higher, if the prescursor 25(OH)D is lower? Does not seem logic.
- The rationale for choosing the intervention period to be in winter time could be explained in the methods section. This could help the reader to better understand the results and interpretation. This is only mentioned at the very end of the discussion, which I find too late in the manuscript.
- l. 317: similar to what? own results or ref. 8?
- l. 376: Spelling: change powdered to powered
- refs. 4 and 15 are the same -> delete one
Reviewer 2 Report
McCourt and O'Sullivan investigated health status and metabolic changes with vitamin D administration. Intervention studies in humans are very valuable. Thus, what to measure from blood samples, how to analyze them, and the data obtained are valuable information. However, what exactly the authors measured is unknown, except for the substances listed in the tables. Since vitamin D is synthesized from 7-dehydrocholesterol, it is worth mentioning the variability of metabolites such as cholesterol precursors and primary bile acids, etc. Vitamin D metabolite, 25-Hydroxyvitamin D, regulates lipid metabolism by inducing degradation of SREBP/SCAP complex. The authors did not mention this and it is not clear if vitamin D metabolites were measured. The influence of vitamin D receptor-mediated signals, such as FGF23 and FGF19, on lipid metabolism should also be mentioned. As the authors state, lipids in the body are kept in balance by homeostasis. It is not clear from the manuscript to what extent the whole picture of lipids, including fatty acids, phospholipids, glycerolipids, and sterols, is measured and analyzed.
Reviewer 3 Report
The hypothesis-generating study of McCourt and O'Sullivan has merits and some findings are indeed insightful from a metabolomics point of view. Despite the novel approach however, the conclusion that vitamin D influences lipid metabolism is well known even in older adults. One caveat that the authors did not mention is that the influence of vitamin D function is sexually dimorphic (Al-Daghri et al, J Proteome Res 2014), and their analysis was never stratified according to sex, probably due to sample size issues. They may need to mention this as a limitation. As for other clarifications:
Major
1. Additional details are needed in describing the participants. Are they community-dwelling elders? Also please provide how much vitamin D content was fortified for each intervention aside from the supplement group. 20mcg is around 800IU D3, please mention this so other readers will grasp the equivalent.
2. A consort flowchart of participants will add clarity to the subject distribution. I believe this is needed even if the main focus was on the metabolomics and not on the trial.
Minor
1. Please define 'habitual' food intake.
2. Page 20, line 376 - 'was not powdered', change to powered
Reviewer 4 Report
Regarding the manuscript "Influence of vitamin D status and supplementation on metabolomic profiles of older adults" submitted to Metabolites journal the following comments should be mentioned:
1. The prospective interventional study was remarkably well conducted, and the article text is clearly and concisely written.
2. However, the following minor issues could be mentioned:
a. Reference 8 is not relevant to the objective of the present study and could be replaced by an adequate reference, in the Introduction chapter.
b. Reference 10 could be replaced also since it underlines the effect of vitamin D in a single population; other relevant studies have shown reduced metabolic disease risk of vitamin D, in the Introduction chapter.
c. The authors state that a total of 32 males and 31 females completed the study, but Table 1 contains data for 73 participants. And then, there are 83 and 75 participants in total in Table 2 and Table 3, respectively. Please, explain.
d. In the Discussion chapter, the authors state that they included healthy older adults. But this fact is not mentioned in the Methods. And if so, how did the authors recruit healthy volunteers only and excluded those suffering from different age-related diseases?
Round 2
Reviewer 2 Report
The authors indicate the substance measured in Supplementary Table 1, but if they are quantified, the quantification values should be stated, and if some of them are below the limit of quantification, they should be described as LOQ.
Author Response
Dear Reviewer,
Thanks again for reviewing our manuscript. We have added metabolite concentrations to Supplementary Table 1 and will upload the revised table to be considered for publication.
Round 3
Reviewer 2 Report
acceptable